# Chloroquine-Based Mitochondrial ATP Inhibitors

**DOI:** 10.3390/molecules28031161

**Published:** 2023-01-24

**Authors:** Zhiguo Wang, Robert J. Sheaff, Syed R. Hussaini

**Affiliations:** Department of Chemistry and Biochemistry, The University of Tulsa, 430 South Gary Place, Tulsa, OK 74104, USA

**Keywords:** chloroquine, hydroxychloroquine, mitochondrial inhibitor

## Abstract

Mitochondria is an important drug target for ailments ranging from neoplastic to neurodegenerative diseases and metabolic diseases. Here, we describe the synthesis of chloroquine analogs and show the results of mitochondrial ATP inhibition testing. The 2,4-dinitrobenzene-based analogs showed concentration-dependent mitochondrial (mito.) ATP inhibition. The most potent mito. ATP inhibitor was found to be *N*-(4-((2,4-Dinitrophenyl)amino)pentyl)-*N*-ethylacetamide (**17**).

## 1. Introduction

Chloroquine and hydroxychloroquine (H)CQ) (Figure 1) have been used in treatment of malaria, rheumatoid arthritis, and lupus erythematosus [1]. According to the most widely accepted mechanism (H)CQ enters the red blood cells through diffusion and then moves into the digestive vacuole (DV) of the *Plasmodium* parasite. The *Plasmodium* forms these DVs for metabolism and survival inside erythrocytes, and these lysosome-like DVs are acidic. The parasite degrades host hemoglobin into amino acids by the action of proteases that operate under an acidic environment. This process generates free heme, which polymerizes into hemozoin under the acidic environment. Free heme is toxic to the parasite, while hemozoin is not. As (H)CQ is a weak base, it increases the pH inside DVs, disrupting proteolysis and formation of hemozoin. Aggregation of cytotoxic heme leads to parasite death [2,3,4,5].

This (H)CQ-mediated pH rise in membrane-bound organelles has been shown to inhibit viruses that require pH-dependent entry [6]. A similar pH-dependent process involving mitochondria (mito.) was later speculated to account for the possible antiviral activities of (H)CQ. In this model, (H)CQ was found to target mito. ATP production and cause a concentration-dependent decrease in ATP levels [7]. Mito. intermembrane space is more than one pH unit lower than the matrix. This difference in pH generates a proton gradient across the membrane, which is partly responsible for mito. ATP production [8]. (H)CQ was proposed to diffuse into mito. intermembrane space, where it acts as a weak base and increases the pH in the intermembrane space, thus impacting ATP production [7]. These findings suggest that (H)CQ and its analogs have the potential to be mito. inhibitors.

To understand what part of (H)CQ is responsible for inhibition of mito. ATP, we studied (H)CQ analogs. Three factors were evaluated. Which part of the (H)CQ causes inhibition of mito. ATP? The alkyl amine were aromatic amine were considered possibilities. Furthermore, is the basic nature of (H)CQ accountable for the inhibition of mito. ATP? We present here our findings.

## 2. Results and Discussion

### 2.1. Synthesis

Three types of compounds were prepared in this study:CQ analogsHCQ analogsA CQ analog with an amide group

We prepared the CQ analog **1** by the S_N_Ar reaction [9] of commercially available amine 2 with 3 (Figure 1). CQ derivatives 4 and 5 were prepared by Buchwald–Hartwig amination [10,11] of 2 with 6 and 7, respectively (Figure 2). The HCQ analog 8 was synthesized using Figure 3. Synthetic intermediates 9 and 10 were prepared by modifying a continuous-flow synthesis method [12]. Reductive amination of 10 gave 11, which was converted into 8 by the S_N_Ar reaction with good yield. The amide analog of CQ was prepared as shown in Figure 4. Amide 12 was synthesized by a modified procedure [13]. Ketalization [14] of 12 gave 13, which upon alkylation formed 14. Deprotection of the ketal 14 gave ketone 15, which was converted into amine 16 via reductive amination. Finally, the S_N_AR reaction between 16 and 3 gave the CQ analog 17 containing an amide group.

### 2.2. Rationale for the Selection of Analogs

(H)CQ has three nitrogen atoms, two of them are basic. The conjugate acid of the tertiary amine has a pka of 10.2, while the pka of the conjugate acid of pyridine nitrogen is 8.2 [4]. All except **17** have the tertiary amine. Due to their structural similarity with (H)CQ, these tertiary amine groups are expected to have pka values similar to (H)CQ. Compounds 1, 4, 5, 8, and 17 also contain the non-basic arylamine nitrogen function. Analog 5 has all three types of nitrogen atoms found in (H)CQ, with a calculated pka value of 6.6 for the conjugate acid of the pyridine nitrogen [15].

We selected the dinitrobenzene analogs of (H)CQ (1, 8, and 17) because, like the chloroquinoline unit of (H)CQ, dinitrobenzene is aromatic. Furthermore, 2,4-dinitrophenol (DNP) is known to block mito. ATP synthesis but permits continued electron transport along the respiratory chain to O_2_. In other words, it is an uncoupler [16]. DNP exists as an anion at intracellular pH. Because of the lower pH near mito., protonation of DNP anion occurs, making it a neutral specie. The neutral specie is more prone to diffuse into and through the membrane into mito. [16]. We therefore expected that the aryl part of 1, 8, and 17 should also enter mito. as it has an even less acidic arylamine function than the phenol in DNP. To test if an aryl portion is necessary for mito. inhibition, fragments 2 and 11 were selected. To deetermine whether the amine function was required for mito. ATP modulation, compound 17 was chosen as it is the same as 1, but with the alkylamine changed to an amide, which is less basic than the amine. Analogs 4 and 5 were selected because they have different aryl groups than 1 and 8, and less acidic arylamine functions (calculated pka values of the arylamine function of 4 and 5 are 20.2 and 21.7, while the pka of the arylamine function for 1 is 12.3 and CQ is 27.3) [15]. Furthermore, 5 contains a pyridine unit also present in (H)CQ. Both 4 and 5 can be prepared from commercially available starting materials in a single step.

### 2.3. Biochemical Data

Two major pathways for ATP production in mito. are cytosolic glycolysis and oxidative phosphorylation [17]. Cells switch to the other pathway for ATP production when one pathway is blocked [7]. Therefore, to test if a compound targets mito., it is necessary to block glycolysis, or the net ATP production from glycolysis must be zero. For this study, we chose the latter option. The net ATP production via glycolysis is zero when cells are grown in galactose instead of glucose. This happens because glycolysis generates two net ATP per molecule of glucose. However, galactose cannot be directly used in cellular metabolism and is first converted into glucose-6-phosphate. This conversion requires two ATP molecules. Therefore, the net generation of ATP by galactose molecules in glycolysis is zero [18].

Two sets of screenings were carried out for each (H)CQ analog (Table 1). In both groups, ATP production was measured in cells treated with different concentrations of an (H)CQ analog. In the first set, the cell culture media was L-15, containing galactose instead of glucose. In the second set, glucose was added to the L-15 media. Mito. ATP inhibition observed in L-15 media should be reversed in the L-15 + glucose media as the glycolysis pathways become available, resulting in net positive ATP production.

Entries 1 and 3 (Table 1) contained aryl groups, while entries 2 and 4 were non-aromatic fragments of (H)CQ. The data suggests that having an alkyl amine group is not sufficient to attain mito. ATP inhibition. However, whereas entry 1 was a potent inhibitor of mito. ATP production, entry 3 modulated ATP production only mildly. A comparison of entries 1 and 3 with 5 and 6 suggests that the nature of the arylamine portion of (H)CQ analogs is also relevant. While entries 1 and 3 seemed to inhibit mito. ATP, 5 and 6 (like entries 2 and 4) were neither potent nor specific in inhibiting mito. ATP production. Finally, the data for entry 7 suggest that the basicity of (H)CQ may also not be the cause of mito. ATP inhibition. Here, the basic function of amine has been converted into an amide, but the analog is even more potent than entry 1 in inhibiting mito. Therefore, we conclude that mito. ATP inhibition by (H)CQ and its analogs occurs via another process. More work is needed to find the mechanism of this mito. ATP inhibition process. Nonetheless, our study identified (H)CQ analogs that are mito. ATP modulators, which as such may find uses in researching pathologies linked with mito. dysfunction [19].

## 3. Conclusions

In this paper, we have described our efforts to target mito. through (H)CQ analogs. We attempted to understand what feature of (H)CQ analogs is responsible for ATP modulation. Neither the alkyl amine nor the basic nature of the (H)CQ analogs alone can account for the mito. modulation. The nature of the arylamine portion of the (H)CQ analogs did have an impact on mito. ATP modulation. Arylamine functions that can deprotonate more easily (such as in 2,4-dinitrobenzene-based analogs) show greater ATP inhibition, and compound 17 was the most potent mito. ATP inhibitor in the series.

## 4. Materials and Methods

### 4.1. Synthesis

#### 4.1.1. General Experimental

Amine (**2**) was purchased from a commercial source. Methanol was dried by standing over on molecular sieves for 3 days. Tetrahydrofuran was dried by distillation from CaH_2_ onto 4 Å molecular sieves. Alumina refers to aluminum oxide 90 active, neutral from EM Reagents. TLCs were visualized using UV or stained with KMnO_4_. The 1H and 13C NMR spectra were performed on a Varian 400/50 (400 MHz) spectrometer and measured in ppm relative to TMS (0.00 ppm) or the residual solvent CHCl_3_ (^1^H NMR *δ* = 7.26 ppm and ^13^C NMR *δ* = 77.16 ppm). High-resolution mass spectrometry measurements were taken using a Thermo Scientific Fusion or Exactive spectrometer operating in a positive ion electrospray mode using an Orbitrap analyzer at a nominal resolution of 120,000. Glass silica gel plates (250 μm) were used for thin-layer chromatography, and sorbent silica gel 60 Å (40–63 μm) was used for flash chromatography. Experimntal procedures for synthesizing **9** and **10**, NMR spectra of isolated compounds, and biochemical data for Table 1 are available in the Appendix A.

#### 4.1.2. *N*-(5-(Diethylamino)pentan-2-yl)-2,4-dinitrobenzenamine

(**1**). A suspension of **3** (98%, 1.93 g, 9.36 mmol) and absolute ethanol (5.90 mL) was heated at 70 °C. When this was entirely dissolved, **2** (97%, 2.06 mL, 10.3 mmol) was added with a syringe pump at a rate of 0.02 mL/min. Afterward, the reaction was refluxed for 4 h. The reaction was cooled to 70 °C and 15.0 M NH_4_OH (15.0 mmol) was added. The reaction was allowed to come to rt and ethanol was removed on a rotary evaporator. Distilled water (15.0 mL) was added to the reaction and the product was extracted with CH_2_Cl_2_ (20.0 mL × 4). The organic phase was dried with Na_2_SO_4_ and evaporated on a rotary evaporator. Vacuum distillation of this crude extract gave pure 1 as a dark brown liquid (2.85 g, 8.79 mmol, 94%). R _f_ = 0.20 (EtOAc, the TLC was pretreated with 1% Et_3_N); 1H NMR (400 MHz, CDCl_3_): δ 9.11 (s, 1H), 8.54–8.53 (m, 1H), 8.26–8.24(m, 1H), 7.04 (d, J = 9.8 Hz, 1H), 3.94–3.82 (m, 1H), 2.69 (q, J = 6.8 Hz, 4H), 2.64–2.54 (m, 2H), 1.94–1.62 (m, 4H), 1.39 (d, J = 6.0 Hz, 3H), 1.12 (t, J = 6.8 Hz, 6H); 13C NMR (100 MHz, CDCl_3_): δ 147.8, 135.7, 130.3, 130.1, 124.5, 114.4, 52.0, 49.1, 46.6, 34.2, 22.9, 20.3, 10.8. HRMS (ESI) *m*/*z*: (M + H) calcd for C_15_H_25_N_4_O_4_, 325.18761; measured 325.18761.

#### 4.1.3. 4-(5-(Diethylamino)pentan-2-ylamino)benzonitrile

(**4**). Pd_2_(dba)_3_ (97%, 2.4 mg, 2.50 μmol), BINAP (97%, 7.8 mg, 7.50 μmol) and sodium *tert*-butoxide (96%, 70.1 mg, 0.700 mmol) were added to an oven-dried pressure vessel in a glovebox. Amines **2** (97%, 99.7 mg, 0.600 mmol) and **6** (99%, 91.9 mg, 0.500 mmol) were transferred to the reaction vessel using anhydrous 1,2-dimethoxyethane (DME) (0.500 mL) under N_2_. Vials containing **2** and **6** were washed with anhydrous 1,2-dimethoxyethane (DME) (2 × 0.250 mL) and the contents were transferred to the reaction vessel under N_2_. The reaction was stirred and heated in an 80 °C oil bath for 24 h. The reaction was allowed to come to rt, evaporated, and flash chromatographed. The flash chromatography column was packed with 1% Et_3_N in EtOAc, and run with EtOAc, then MeOH in EtOAc (10%, then 20%, then 30%), giving a yellow glue-like material. This material was treated with NaOH (1.00 M, 5.00 mL) and transferred to a separatory funnel. The container was washed with CH_2_Cl_2_ (1.70 mL × 3) and the contents were transferred to the separatory funnel and extracted with CH_2_Cl_2_ (5.00 mL × 3). The combined organic layer was dried with Na_2_SO_4_ and evaporated to provide pure **4** as a dark orange oil (119 mg, 0.46 mmol, 92%). R*_f_* = 0.21 (3:7 MeOH/EtOAc, the TLC was pretreated with 1% Et_3_N); 1H NMR (400 MHz, CDCl_3_): *δ* 7.38 (d, *J* = 8.6 Hz, 2H), 6.51 (d, *J* = 8.6 Hz, 2H), 4.68 (br, 1H), 3.51 (br, 1H), 2.56–2.47 (m, 4H), 2.43–2.40 (m, 2H), 1.62–1.51 (m, 4H), 1.20 (d, *J* = 6.0 Hz, 3H), 1.01 (t, *J* = 7.2 Hz, 6H); 13C NMR (100 MHz, CDCl_3_): *δ* 150.9, 133.8, 120.8, 112.3, 97.6, 52.8, 48.0, 46.7, 34.8, 23.6, 20.4, 11.5. HRMS (ESI) *m*/*z*: (M + H) calcd for C_16_H_26_N_3_, 260.21267; measured 260.21261.

#### 4.1.4. *N*-(5-(Diethylamino)pentan-2-yl)pyridin-2-amine

(**5**) [20]. Pd(OAc)_2_ (98%, 1.1 mg, 5.00 µmol), Josiphos (97%, 4.3 mg, 7.50 μmol), and sodium tert-butoxide (96%, 70.1 mg, 700 mmol) were added to an oven-dried pressure vessel in a glovebox. Amines **2** (97%, 120 μL, 0.600 mmol) and 7 (99%, 57.3 mg, 0.500 mmol) were transferred to the reaction vessel using anhydrous 1,2-dimethoxyethane (DME) (0.500 mL) under N_2_. Vials containing **2** and **7** were washed with anhydrous 1,2-dimethoxyethane (DME) (2 × 0.250 mL) and the contents were transferred to the reaction vessel under N_2_. The reaction was stirred and heated in an 80 °C oil bath for 24 h. The reaction was allowed to come to rt, evaporated, and flash chromatographed. The flash chromatography column was packed with 1% Et_3_N in EtOAc, and run with EtOAc, then MeOH in EtOAc (10%, then 20%, then 30%), giving a brown glue-like material. This material was treated with NaOH (1.00 M, 5.00 mL) and transferred to a separatory funnel. The container originally holding the evaporated column fractions was washed with CH_2_Cl_2_ (1.70 mL × 3) and the contents were transferred to the separatory funnel and extracted with CH_2_Cl_2_ (5.00 mL × 3). The combined organic layer was dried with Na_2_SO_4_ and evaporated to give pure 5 as a dark orange oil (116 mg, 0.49 mmol, 99%). R_f_ = 0.30 (2:8 MeOH/EtOAc, the TLC was pretreated with 1% Et_3_N); 1H NMR (400 MHz, CDCl_3_): δ 8.06–8.05 (m, 1H), 7.39–7.35 (m, 1H), 6.52–6.49 (m, 1H), 6.34 (d, J = 8.4 Hz, 1H), 4.51 (s, br, 1H), 3.85–3.71 (m, 1H), 2.51 (q, J = 7.0 Hz, 4H), 2.45–2.40 (m, 2H), 1.59–1.47 (m, 4H), 1.20 (d, J = 6.8 Hz, 3H), 1.00 (t, J = 7.0 Hz, 6H); 13C NMR (100 MHz, CDCl_3_): δ 158.4, 148.3, 137.3, 112.3, 106.8, 52.9, 47.1, 46.8, 35.2, 23.7, 21.0, 11.7. (ESI) *m*/*z*: (M + H) calcd for C_14_H_26_N_3_, 236.21267; measured 236.21267.

#### 4.1.5. (4-. Aminopentyl)(ethyl)amino)methanol

(**11**). Acetic acid (1.23 mL, 21.3 mmol) was added to a stirred mixture of compound **10** (1.23 g, 7.08 mmol) and NH_3_ (7.00 M in MeOH, 8.30 mL) under N_2_. Nitrogen was added through a needle passing through a rubber septum. The N_2_ line was removed after the addition to minimize the loss of NH_3_, and the mixture was stirred for 30 min. Sodium cyanoborohydride (0.703 g, 10.6 mmol) was dissolved in NH_3_ (7.00 M in MeOH, 4.00 mL) and added to the reaction mixture under N_2_. The N_2_ line was removed again and the reaction mixture was stirred. After stirring for 48 h at rt, the solvent was removed and aqueous NaOH (3.0 M, 15 mL) was added to the residue. The crude product was extracted with CH_2_Cl_2_ (15 mL × 4), dried over Na_2_SO_4_, and evaporated to give semi-pure **11** as a yellow oil (996 mg, 81%). The semi-pure 11 was satisfactory for the next step reaction, but highly pure **11** was required for biological tests. Further purification by vacuum distillation provided highly pure **11** as a colorless oil (mass of semi-pure = 0.652 mg, mass of pure = 0.566 mg, 87%). The 1 H and 13 C spectra matched those reported [12]: 1H NMR (400 MHz, CDCl_3_): δ 3.54 (t, J = 5.5 Hz, 2H), 2.89 (tq, J = 6.4, 6.4 Hz, 1H), 2.58 (t, J = 5.6 Hz, 2H), 2.55 (t, J = 7.0 Hz, 2H), 2.46 (t, J = 7.4 Hz, 2H), 2.10 (s, br, 2H), 1.56–1.40 (m, 2H), 1.36–1.27 (m, 2H), 1.07 (d, J = 6.4 Hz, 3H), 1.02 (t, J = 7.4 Hz, 3H); 13C NMR (100 MHz, CDCl_3_): δ 58.5, 55.1, 53.4, 47.3, 46.9, 37.9, 24.2, 24.1, 11.8.

#### 4.1.6. 2-((4-((2,4-Dinitrophenyl)amino)pentyl)(ethyl)amino)ethanol

(**8**). Compound **3** (98%, 43.3 mg, 0.209 mmol) was dissolved in absolute ethanol (0.210 mL) at 65 °C, then **11** (40.1 mg, 0.230 mmol) was added to this clear solution dropwise and the reaction refluxed overnight. Ammonium hydroxide (15.0 M, 35.0 μL) was added to the reaction at rt. The solvent was evaporated and distilled water (2.00 mL) was added to the reaction mixture. The product was extracted with CH_2_Cl_2_ (4.00 mL × 3), dried over Na_2_SO_4_, and evaporated to give pure 8 as a brown liquid (57.7 mg, 81%). Spectral values were 1H NMR (400 MHz, CDCl_3_): δ 9.14 (s, 1H), 8.53 (d, J = 7.2 Hz, 1H), 8.25 (d, J = 10.0 Hz, 1H), 6.98 (d, J = 10.0 Hz, 1H), 4.51 (s, br, 1H), 3.87–3.80 (m, 1H), 3.65 (t, J = 5.0 Hz, 2H), 2.76–2.68 (m, 6H), 1.74–1.69 (m, 4H), 1.39 (d, J = 6.4 Hz, 3H), 1.11 (t, J = 7.2 Hz, 3H); 13C NMR (100 MHz, CDCl_3_): δ 147.8, 135.7, 130.4, 130.2, 124.6, 114.3, 58.0, 55.2, 52.9, 49.2, 47.7, 34.1, 23.1, 20.4, 11.0. (ESI) *m*/*z*: (M + H) calcd for C_15_H_25_N_4_O_5_, 341.18253; found *m*/*z* 341.18230.

#### 4.1.7. *N*-(4-Oxopentyl)acetamide

(**12**) [13]. A solution of acetyl chloride (0.355 mL, 4.94 mmol) in CH_2_Cl_2_ (5.00 mL) was added dropwise to a well-stirred solution of pyridine (0.800 mL, 9.79 mmol) in CH_2_Cl_2_ (10. mL) under N_2_. The mixture was stirred for 15 min. Then the solution of 2-methyl-1-pyrroline (0.488 mL, 4.90 mmol) in CH_2_Cl_2_ (12.5 mL) was added dropwise into this mixture. The mixture was stirred until it became clear orange, then it was stirred for another 10 min. Next, HCl (5%, 10.0 mL) was added to the mixture under vigorous stirring, and the mixture was continually stirred for 15 min. Sodium hydroxide solution (10.0 M, 3.00 mL) was added to the reaction mixture, followed by 10 min. of stirring. The organic layer was separated and the aqueous layer was extracted with CH_2_Cl_2_ (15.0 mL × 3). The combined organic phase was dried over Na_2_SO_4_, evaporated, and flash-chromatographed. The flash-chromatography column was packed with 1% Et_3_N in EtOAc and was run with EtOAc, then 8% MeOH in EtOAc, giving **12** as a white solid (434 mg, 3.03 mmol, 62%). R*_f_* = 0.30 (1:9 MeOH/EtOAc). The 1H and 13C spectra matched the reported values [13].

#### 4.1.8. *N*-(4,4-Dimethoxypentyl)acetamide

(**13**). This procedure is based on a general method reported for ketalization [14]. Trimethyl orthoformate (1.68 mL, 15.0 mmol), dry methanol (40.0 mL), and concentrated sulfuric acid (14.3 μL) were added to a round bottom flask containing **12** (0.430 g, 3.00 mmol) under N_2_. The reaction was stirred for 72 h. Next, NaHCO_3_ (86.0 mg) was added and stirred for 30 min. The solvent was removed and the residue was dissolved in EtOAc and filtered through a piece of cotton. The solvent was removed to give pure **13** as a brown-orange oil (568 mg, 100%). Spectral values were 1H NMR (400 MHz, CDCl_3_, basified with Na_2_CO_3_): *δ* 5.96 (s, br, 1H), 3.25 (dt, *J* = 6.0, 7.2 Hz, 2H), 3.17 (s, 6H), 1.97 (s, 3H), 1.65–1.51 (m, 4H), 1.26 (s, 3H); 13C NMR (100 MHz, CDCl_3_): *δ* 170.2, 101.4, 48.1, 39.8, 33.9, 24.5, 23.3, 21.0. (ESI) *m*/*z*: (M + H) calcd for C_9_H_19_NO_3_, 212.12626; found *m*/*z* 212.12648.

#### 4.1.9. *N*-(4,4-Dimethoxypentyl)-*N*-ethylacetamide

(**14**). A solution of **13** (0.539 g, 2.85 mmol) in dry THF (3.00 mL) was added to a flask containing NaH (60% dispersion in mineral oil, 0.342 g, 8.55 mmol) under N_2_. The vial containing **13** was washed (2 × 1.00 mL) of THF and the contents were transferred to the reaction flask. The mixture was stirred and heated at 30 °C for 1 h. Iodoethane (0.917 mg, 5.70 mmol) was dissolved in THF (0.500 mL) and added to the reaction mixture under N_2_. The vial containing iodoethane was washed (2 × 0.250 mL) and the contents were transferred to the reaction flask. The reaction was heated at 30 °C for an additional 1 h. Next, the reaction was refluxed for 24 h under N_2_, using a condenser chilled at 0 °C by a circulating coolant. Finally, the reaction was allowed to come to rt and the solvent was evaporated. The residue was dissolved in Et_2_O, filtered through a cotton plug, and passed through a short-path alumina column. After solvent removal, **14** was obtained as a reddish-orange liquid mixture of rotamers (1.4:1) (619 mg, 2.85 mmol, 100%). Spectral values were 1H NMR (400 MHz, CDCl_3_): *δ* 3.41–3.23 (m, 4H), 3.18 (s, 6H (minor)), 3.17 (s, 6H (major)), 2.08 (s, 3H), 1.97 (s, 3H), 1.60–1.58 (m, 4H), 1.28 (s, 3H (minor), 1.26 (s, 3H (major)), 1.18 (t, *J* = 7.2 Hz, 3H (major)), 1.12 ((t, *J* = 7.2 Hz, 3H (minor)); 13C NMR (100 MHz, CDCl_3_): *δ* (major) 170.0, 101.5, 48.19, 48.15, 43.4, 33.8, 24.0, 21.7, 21.2, 13.1; (minor) 169.9, 101.3, 48.5, 45.5, 40.5, 33.7, 22.8, 21.5, 21.1, 14.2. (ESI) *m*/*z*: (M + H) calcd for C_11_H_23_NO_3_Na, 240.15756; found *m*/*z* 240.15773.

#### 4.1.10. *N*-Ethyl-*N*-(4-oxopentylacetamide

(**15**). Glacial acetic acid (9.53 mL), 1.00 M HCl (4.77 mL), and THF (14.3 mL) were added to a flask containing **14** (0.589 g, 2.71 mmol), and the reaction was stirred for 24 h. The organic solvents were evaporated and NaOH solution (16.7 M, 12.0 mL) was added dropwise into the reaction vessel at 0 °C. The product was extracted with CH_2_Cl_2_ (30 mL × 4), dried over Na_2_SO_4_, and evaporated to obtain crude **15** as a dark yellow oil. Purification by flash chromatography (100% EtOAc, then 1% MeOH in EtOAc) yielded pure **15** as a mixture of rotamers (yellow oil, 242 mg, 1.41 mmol, 52%) in a ratio of 1.7:1. R *_f_* = 0.18 (1:9 MeOH/EtOAc). Spectral values for 1H NMR (400 MHz, CDCl_3_): *δ* 3.40–3.21 (m, 4H), 2.50–2.448 (m, 2H), 2.17 (s, 3H (minor)), 2.14 (s, 3H (major)), 2.10 (s, 3H (minor)), 2.08 (s, 3H (major)), 1.87–1.77 (m, 2H), 1.18 (t, *J* = 7.0 Hz, 3H (major)), 1.11 (t, *J* = 7.2 Hz, 3H (minor)); 13C NMR (100 MHz, CDCl_3_): *δ* (major) 208.4, 170.3, 44.1, 40.7, 39.9, 30.0, 21.8, 21.4, 14.0; (minor) 207.4, 170.0, 47.4, 43.1, 40.3, 30.1, 22.6, 21.5, 13.0. (ESI) *m*/*z*: (M + H) calcd for C_9_H_18_NO_2_Na, 172.13376; found *m*/*z* 172.13351.

#### 4.1.11. *N*-(4-Aminopentyl)-*N*-ethylacetamide

(**16**). Acetic acid (0.234 mL, 4.20 mmol) was added to a stirred mixture of **15** (0.240 g, 1.40 mmol) and NH_3_ (7.00 M in MeOH, 1.63 mL) under N_2_. Nitrogen was added via a needle passing through a rubber septum. After the addition, the N_2_ line was removed to minimize the loss of NH_3_ and the mixture was stirred for 30 min. Sodium cyanoborohydride (0.139 g, 2.10 mmol) was dissolved in NH_3_ (7.00 M in MeOH, 0.800 mL) and added to the reaction mixture under N_2_. The N_2_ line was removed again and the reaction mixture was stirred. After 24 h, the solvent was removed and aqueous NaOH (4.00 M, 4.00 mL) was added to the residue. The crude product was extracted with CH_2_Cl_2_ (5 mL × 4), dried over Na_2_SO_4_, and evaporated to give **16** as a yellow oil (240 mg, 99%). The product was obtained as a 5:1 mixture of rotamers. Values for the 1H NMR spectrum (400 MHz, CDCl_3_): *δ* 3.40–3.22 (m, 4H), 2.96–2.86 (m, 1H (major)), 2.74–2.68 (m, 1H (minor)), 2.09 (s, 3H (major)), 2.08 (s, 3H (minor)), 1.90–1.48 (m, 4H), 1.39–1.26 (m, 2H), 1.18 (t, *J* = 7.0 Hz, 3H (major)), 1.11 (t, *J* = 7.0 Hz, 3H (minor)), 1.08–1.05 (m, 3H); 13C NMR (100 MHz, CDCl_3_): *δ* (major) 169.9, 48.5, 45.3, 43.3, 37.1, 26.8, 24.1, 14.1; (minor) 170.0, 46.79, 46.77, 40.4, 37.3, 26.0, 24.3, 21.5, 13.0. (ESI) *m*/*z*: (M + H) calcd for C_9_H_21_N_2_O, 173.16539; found *m*/*z* 173.16498.

#### 4.1.12. *N*-(4-((2,4-Dinitrophenyl)amino)pentyl)-*N*-ethylacetamide

(**17**). Potassium carbonate (98%, 41.9 mg, 0.300 mmol), **3** (99%, 62.0 mg, 0.300 mmol), Et_3_N (41.8 μL, 0.300 mmol), **16** (62.0 mg, 0.360 mmol), and absolute ethanol (0.300 mL) were heated and stirred at 85 °C for 16 h. The reaction was allowed to come to rt and the solvent was removed on a rotary evaporator. The crude extract was purified by flash chromatography. The column was packed with 1:25:74 Et_3_N/EtOAc/hexanes and rinsed with 1:3 EtOAc/hexanes (350 mL). The column was then loaded with the crude extract and eluted with 9:1 EtOAc/hexanes, giving **17** (84.0 mg, 0.250 mmol, 83%) as a red oil. The product was obtained as a 3:1 mixture of rotamers. R *_f_* = 0.22 (EtOAc, the TLC was pretreated with 1% Et_3_N in EtOAc). Spectral values were 1H NMR (400 MHz, CDCl_3_): *δ* 9.14–9.12 (m, 1H), 8.56–8.46 (m, 1H), 8.29–8.24 (m, 1H), 6.98 (d, *J* = 10.0 Hz, 1H (major)), 6.93 (d, *J* = 9.6 Hz, 1H (minor)), 1.77–1.62 (m, 4H), 1.40 (d, *J* = 6.4 Hz, 3H (minor)), 1.36 (d, *J* = 6.4 Hz, 3H (major)), 1.21–1.17 (m, 3H (major)), 1.13–1.10 (s, 3H (minor)); 13C NMR (100 MHz, CDCl_3_): *δ* (major) 170.4, 147.9, 135.8, 130.5, 124.66, 114.3, 49.0, 44.4, 43.3, 33.8, 24.4, 21.5, 20.6, 14.1 (minor) 169.8, 147.7, 130.6, 130.2, 124.69, 114.0, 49.2, 48.0, 40.5, 33.9, 25.7, 21.8, 20.6, 13.1. HRMS (ESI) *m*/*z*: (M + H) calcd for C_15_H_23_N_4_O_5_, 339.16688; measured 339.16673.

### 4.2. Bioassay

H293 kidney epithelial cells were obtained from the American Tissue Culture Collection. Cells were maintained in Dulbecco’s Modified Eagle Medium (DMEM) supplemented with 10% fetal bovine serum (FBS) plus penn/strep. Cultures were maintained in a 37 °C water-jacketed incubator with 5% CO_2_. When needed, cells were removed from the plate using trypsin, gently pelleted by centrifugation (2 rpm/min. at room temperature), aspirated, and re-suspended in the desired media. Unless noted otherwise, FBS was omitted because experiments showed it was not required for these short-term metabolic assays (data not shown). The other media used was L-15, which contained amino acids but replaced glucose with 5 mM galactose. L-15 also lacked NaHCO_3_, which was added to the same concentration as that in DMEM (44 mM). Its presence is essential to buffer the media and prevent acidification in the CO_2_ incubator.

In general, the immediate short-term effects of the indicated (H)CQ analogs on ATP levels were analyzed by removing growing cells from a p100mm plate using trypsin, washing with PBS, and re-suspending in the indicated media. After counting using a hemocytometer, 25–50 k cells in 100 microliters of media were distributed in a 96-well white flat-bottom non-treated polystyrene assay plate. The plate was typically incubated for 30 min. at 37 °C in a CO_2_ incubator to allow metabolic pathways to adapt to the media. Different concentrations of the drug of interest were then added, followed by shaking (700 RPM/10 sec), and the cells returned to the incubator. Incubation times were generally between 1 and 2 h so that the direct and immediate effects on ATP production could be evaluated. ATP levels were determined by adding 10 microliters of CellTiterGlo (CTG) reagent directly to the wells, followed by 5 min. shaking at 700 RPM. The light emitted by the ATP-dependent luciferase was quantitated using a photo-luminometer. Samples were tested in duplicate and the standard deviation was determined.

In some cases, the longer-term effects of the drugs were determined by seeding cells in tissue-culture-treated 96-well plates so that they could attach overnight, then treating them with the drug for longer times (e.g., 18 h). Drug effects on ATP were determined using the CTG described above, while viability was measured by the addition of 10 microliters of CellTiterBlue resazurin reagent. In the latter case, cells were incubated for 2 h at 37 °C in the CO_2_ incubator to allow viable cells to convert resazurin to fluorescent resorufin. Fluorescence (ex^5^60/em590) was measured on a plate reader. In all cases, drug effects on the biological assays were evaluated in the absence of cells to ensure they were not targeting assay components or reactions.

## Data Availability

Data presented in this study are available in the paper and the Appendix A.

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
