# Peer review of "Chloroquine-Based Mitochondrial ATP Inhibitors"

_molecules, 2023, doi:10.3390/molecules28031161_

Round 1

Author Response

Please find our responses in the attached file. 

Reviewer 2 Report

This MS is on the synthesis and potential application of Chloroquine and hydroxychloroquine ((H)CQ) analogs as inhibitors of mitochondrial ATP production. The main hypothesis is that the compounds selected would have an effect because of their structural similarity with CQ and HCQ, but the authors also want to check whether their basicity plays a major role in the process by increasing the pH value in the system. For this purpose, a series of compounds had been synthesized and tested. The rationale for the fragment selection sounds, but the reviewer is wondering why compounds with 4-pyridinyl substituent are not studied. Furthermore, given the easy synthesis, the reviewer is wondering why only a limited number of compounds had been obtained and tested. This would allow better SAR to be established. A comparative analysis towards standards (CQ and HCQ) are also missing. 

Some minor remarks are as follows

1)            The synthesis of compound 17 is missing from the script.

2)            The acidity/basicity is the pillar of this study, but no discussion/comparative analysis on these properties for the standard compounds CQ and HCQ is given. 

  Overall, this MS presents a study, which puts forward a hypothesis that is rejected, thus opening up horizons for future investigation. The reviewer's opinion is that it can be published as a preliminary study/short communication if the journal policy allows this

Author Response

(The authors gave the same response as above.)

Round 2

Reviewer 1 Report

The authors have made the indicated corrections. The manuscript has been successfully improved

Author Response

There are no corrections or suggestions by reviewers.